# Enabling three-dimensional porous architectures via carbonyl functionalization and molecular-specific organic-SERS platforms

Ibrahim Deneme [1,3], Gorkem Liman[2,3], Ayse Can[1], Gokhan Demirel [2✉] & Hakan Usta [1✉]

Molecular engineering via functionalization has been a great tool to tune noncovalent intermolecular interactions. Herein, we demonstrate three-dimensional highly crystalline nanostructured **D(C₇CO)-BTBT** films via carbonyl-functionalization of a fused thienoacene π-system, and strong Raman signal enhancements in Surface-Enhanced Raman Spectroscopy (SERS) are realized. The small molecule could be prepared on the gram scale with a facile synthesis-purification. In the engineered films, polar functionalization induces favorable out-of-plane crystal growth via zigzag motif of dipolar C = O···C = O interactions and hydrogen bonds, and strengthens π-interactions. A unique two-stage film growth behavior is identified with an edge-on-to-face-on molecular orientation transition driven by hydrophobicity. The analysis of the electronic structures and the ratio of the anti-Stokes/Stokes SERS signals suggests that the π-extended/stabilized LUMOs with varied crystalline face-on orientations provide the key properties in the chemical enhancement mechanism. A molecule-specific Raman signal enhancement is also demonstrated on a high-LUMO organic platform. Our results demonstrate a promising guidance towards realizing low-cost SERS-active semiconducting materials, increasing structural versatility of organic-SERS platforms, and advancing molecule-specific sensing via molecular engineering.

[1] Department of Nanotechnology Engineering, Abdullah Gül University, 38080 Kayseri, Turkey. [2] Bio-inspired Materials Research Laboratory (BIMREL), Department of Chemistry, Gazi University, 06500 Ankara, Turkey. [3]These authors contributed equally: Ibrahim Deneme, Gorkem Liman. ✉email: nanobiotechnology@gmail.com; hakan.usta@agu.edu.tr

Noncovalent intermolecular interactions play an exceptional role in nature to define the supramolecular arrangement of organic structures, and to govern chemical reactions and different states of matter[1–3]. Although the double-helix structure of DNA and protein secondary structures are formed via multiple strong hydrogen bonds, carbonyl···carbonyl interactions as well as dipolar/London dispersion forces determine three-dimensional structures and functions of these biomaterials[4–7]. Molecular recognition in biological processes (e.g., between and within peptides, proteins, and enzyme-inhibitor complexes) also rely on varied noncovalent interactions[1]. With the advent of synthesis methodologies and characterization tools, varied π-units, heteroatoms, functional groups, and substituents have been employed in the rational design of (semi)synthetic organic structures, which has allowed for the precise control of noncovalent intermolecular interactions in the solid-state to achieve desired supramolecular arrangements, microstructures, and morphologies in various nanotechnology fields. In this perspective, among carbon-based material families developed in the recent decades, semiconducting small molecules have become a key player in thin-film optoelectronic devices[8,9]. Their ability to form nano-/micro-structured thin-films from solution or vapor phase with tunable morphologies and optoelectronic properties has greatly benefited the research for next-generation optoelectronic devices[10–12]. Although most of these device technologies, such as organic field-effect transistors (OFETs)[13–15] or light-emitting diodes[16], have now become a conventional application avenue for molecular semiconductors, they hold huge promise also for unconventional applications such as surface-enhanced Raman spectroscopy (SERS). As we have recently disclosed in our pioneering studies[17,18], the nanostructured organic films of π-electron-deficient fluorinated oligothiophene semiconductors **DFH-4T**[17] and **DFP-4T**[18], fabricated via physical vapor deposition (PVD), enabled the Raman detection of organic analytes (e.g., methylene blue (MB) and rhodamine 6 G) without needing a metallic or an inorganic layer. Our initial findings have convincingly demonstrated that low-lowest unoccupied molecular orbital (LUMO) oligothiophene semiconductors in their nanostructured films could enhance Raman signals via a chemical enhancement mechanism. In this perspective, perfluoro-alkyl-/-aryl substituents attached directly to the quaterthiophene π-core do not only lower the LUMO energy of the π-system (−3.2 to −3.4 eV) for effective charge-transfer (CT) interactions with analyte molecules but also enable a high favorable crystal growth dynamics during the PVD processes. To this end, from a molecular engineering perspective, non-fluorinated molecular structures and π-frameworks other than oligothiophenes are yet to be studied. Exploring additional molecular structures is important not only to widen the scope of SERS-active organic films for potential molecule-specific sensing but also to reveal key molecular parameters that control Raman enhancement mechanisms.

In search of π-frameworks as the building blocks of organic-SERS platforms, [1]benzothieno[3,2-b][1]benzothiophenes (**BTBTs**) stand out as an attractive molecular family. **BTBTs** have been one of the most attractive semiconductor families of this decade[19,20]. This is mainly a result of their favorable charge-transport characteristics in thin films, as well as structural and functional properties, including facile synthesis/functionalization in a small number of steps, solubility in common organic solvents, optical transparency, and convenient film preparation via vacuum- or solution-processing[21,22]. As a result of their completely coplanar electron-rich fused thienoacene π-system, and highly delocalized frontier orbital wave functions with polarizable S-atoms, **BTBTs** are generally hole-transporting (p-type) semiconductors and have convincingly demonstrated high hole mobilities of >1.0–2.0 cm$^2$/Vs in OFETs for a wide range of alkyl and

aryl substituents[23]. Despite their great properties, **BTBTs** had not been preferred as organic-SERS platforms in our previous studies since they are quite π-electron-rich systems with high-lying LUMOs (>−2.2 eV). In the only reported SERS study based on a **BTBT** derivative[24], we have fabricated 2,7-dioctyl[1]benzothieno[3,2-b][1] benzothiophene (**C$_8$-BTBT**) film as a micro-structured organic template to deposit Au nanoparticles for plasmonic Raman enhancement. As the research to tailor **BTBT** π-framework continues in the optoelectronics field, we have recently developed highly π-electron-deficient **BTBTs** via 2,7-functionalizations[25,26]. Our study demonstrated that carbonyl functionalities could adopt in-plane conformations ($\theta_{torsions} < 2–3°$) with the BTBT π-system, which extends and stabilizes ($\Delta E_{LUMO} = −1.5$ eV) LUMO significantly leading to the first examples of electron-transporting (n-type, electron mobility ~0.6 cm$^2$/V s) **BTBT** semiconductors in the literature. Furthermore, carbonyl functionalities were observed to induce strong dipolar and π-interactions between the **BTBTs**, leading to enhanced cohesive energetics relative to non-functionalized analogs[26,27]. These results have not only revealed the great potential of functionalized low-LUMO **BTBTs** in n-type OFETs and complementary circuits, but also made this thienoacene molecular family, as pure organic films, quite attractive for nanostructured SERS-active platforms.

Herein, we explore the nanostructured film formation and Raman signal enhancement abilities of a π-electron-deficient low-LUMO **BTBT** molecule, 1,10-(benzo[b]benzo[4,5]thieno[2,3-d] thiophene-2,7-diyl)bis(octan-1-one) (**D(C$_7$CO)-BTBT**, Fig. 1a), which includes 2,7-dicarbonyl functionalization along with n-heptyl (-n-C$_7$H$_{15}$) substituents. This molecule is prepared on the gram scale in ambient via facile Friedel-Crafts acylation and precipitation/solvent washing without requiring any high-cost transition-metal catalyst and tedious chromatographic/sublimation-based purification. Although the good chemical/thermal stabilities allow for stable semiconductor film deposition, carbonyl functionalization induces favorable out-of-plane crystal growth via zigzag motif of dipolar C = O···C = O interactions, hydrogen bonds, and strengthened π-interactions. A unique two-stage film growth behavior is identified by a transition from an initial densely packed 2D island-based (Volmer-Weber (VW)) morphology into a highly porous 3D surface consisting of vertically oriented nanoplates. This transition is accompanied by an "edge-on" to "face-on" molecular orientational switch driven by the surface hydrophobicity of the initially formed island-based morphology. A non-functionalized BTBT analog, **C$_8$-BTBT**, provides a straight comparison at the molecular level to study the effects of carbonyl functionalization and stabilized/extended frontier π-orbitals on nanostructured film formation and Raman signal enhancements. Three-dimensional highly crystalline nanostructured **D(C$_7$CO)-BTBT** films showed strong Raman signal enhancements in SERS with four different analyte molecules having varied electronic structures. Molecular sensitivity for Raman signal enhancement is also achieved on the non-functionalized (high-LUMO) organic-SERS platform. The analysis of the electronic structures and the ratio of the anti-Stokes to Stokes SERS, suggests that the π-extended and stabilized LUMOs with crystalline face-on orientations in varied directions, all of which are the direct results of carbonyl functionalization, are key to the realization of strong chemical enhancement mechanism.

## Results
**Synthesis and purification of D(C$_7$CO)-BTBT.** As shown in Fig. 1, gram-scale synthesis of **D(C$_7$CO)-BTBT** was carried out in one step from the **BTBT** π-core via Friedel-Crafts acylation following a slightly modified procedure from the literature (60–70% yield)[23,28]. Most importantly, this reaction was performed in ambient without requiring an inert atmosphere although the yield

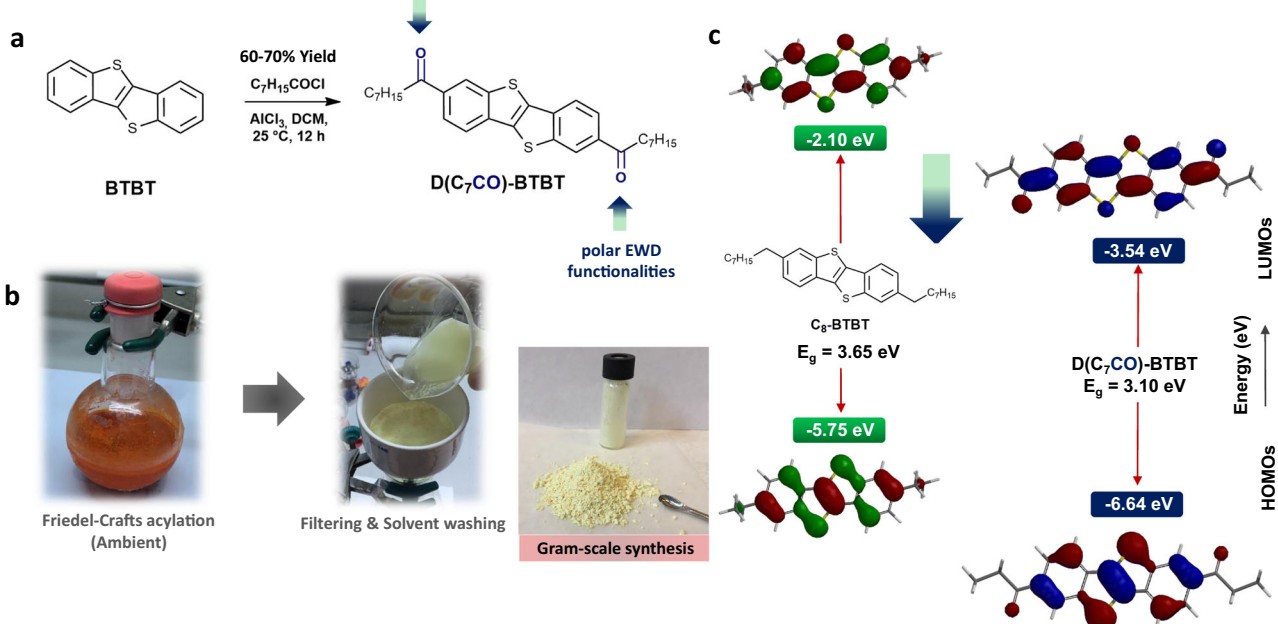

**Fig. 1 Synthesis and purification of D(C₇CO)-BTBT, and computational analysis. a** Chemical structure and synthesis of **D(C₇CO)-BTBT** from the BTBT π-core via Friedel-Crafts acylation reaction in ambient. **b** Photos taken during the ambient reaction and the simple work-up (filtering and washing with water/methanol/dichloromethane), and the photo of the gram-scale solid obtained after drying. **c** Chemical structure of **C₈-BTBT**, experimental HOMO/LUMO energy levels of **C₈-BTBT** and **D(C₇CO)-BTBT** determined via electrochemical methods combined with the optical band gaps ($E_g$'s)[26], and the DFT(B3LYP/ 6-31 G**)-calculated topographical frontier orbital representations.

is ~8% lower than the reaction performed under nitrogen. The target compound was precipitated from the reaction mixture as a result of its low solubility in common organic solvents, which was filtered and purified via simple solvent washing to yield pure **D(C₇CO)-BTBT** solid (Fig. 1b). Therefore, tedious chromatographic purification or thermal gradient sublimation, as typically used in organic semiconductors including our previous SERS-active molecules **DFH-4T** and **DFP-4T**, were not needed. The chemical structure and purity of **D(C₇CO)-BTBT** solid were characterized and established by using $^1H/^{13}C$-nuclear magnetic resonance (NMR) spectroscopy (Supplementary Figs. 1 and 2), mass spectrometry (Supplementary Fig. 3), thin-layer chromatography, and elemental analysis. This solid was directly used for the PVD fabrication of nanostructured SERS-active organic films. The synthesis of **D(C₇CO)-BTBT** did not require any high-cost transition-metal catalyst as used in the synthesis of our previously reported SERS-active molecules **DFH-4T**[17] and **DFP-4T**[18] (i.e., Pd(PPh₃)₄ and Pd(PPh₃)₂Cl₂). Instead, a relatively low-cost AlCl₃ Lewis acid catalyst was used. From a functional materials development standpoint with an aim for low-cost sensing/ detection, **D(C₇CO)-BTBT** stands out as a promising material that could be industrially scalable at low production costs. On the other hand, **C₈-BTBT** was synthesized in accordance with the reported procedure[23]. The only structural difference between **C₈-BTBT** and **D(C₇CO)-BTBT** is the electron-withdrawing (EWD) carbonyl units, and this comparative study at the molecular level provides a solid understanding of the effects of carbonyl functionalization and stabilized/extended frontier π-orbitals on nanostructured film formation and SERS ability of molecular π-systems.

**Fabrication of nanostructured D(C₇CO)-BTBT and C₈-BTBT films.** The micron-thick films of **D(C₇CO)-BTBT** (~5.5 ± 0.2 μm thick) and **C₈-BTBT** (~2.3 ± 0.2 μm thick) were deposited onto Si(001) substrates under high vacuum ($1 \times 10^{-6}$ Torr) via a modified PVD technique (Fig. 2a). Different than conventional

slow thermal evaporation process used for the fabrication of layer-by-layer grown semiconducting thin-films (≤100 nm) in organic optoelectronics, the formation of out-of-plane oriented crystalline nanostructures in a micron-thick film was enabled by employing modified parameters of fast molecular deposition rate (~40 nm/s), short source-to-substrate distance (~7 cm), and low substrate temperature (~25–30 °C). This film deposition condition enables ballistic molecular transport from the source-to-substrate surface and promotes the formation of out-of-plane morphologies by minimizing in-plane crystal growth[8,29]. However, using a modified PVD technique does not always guarantee the desired morphology in organic films; the characteristics of intermolecular interactions in the out-of-plane direction are also crucial, which is strongly correlated with the molecular structure and functional units.

**Microstructures, morphologies, and film growth mechanisms of nanostructured D(C₇CO)-BTBT and C₈-BTBT films.** The semiconductor film morphologies were characterized using field emission scanning electron microscopy (FE-SEM) (Fig. 2). The top-view FE-SEM analysis of **D(C₇CO)-BTBT** (Fig. 2f) film showed uniformly distributed and highly interconnected vertically oriented nanoplates with lateral sizes of ~50–70 nm. On the other hand, as shown in Fig. 2c, the **C₈-BTBT** film showed a significant amount of face-on-oriented 2D plate formation with a limited density of vertical plates (~100–200 nm lateral sizes). When compared with **C₈-BTBT**, the surface of **D(C₇CO)-BTBT** film showed a much higher density of vertically oriented thin nanoplates with an extensive porosity. The wettability of these films was studied by water contact-angle measurements (Fig. 2b and e insets), which showed hydrophobic surfaces (CA$_{water}$ ≥ 90°) for both films with a much larger contact angle of 142.5 ± 5.3° for **D(C₇CO)-BTBT**. Despite its highly polar nature[30–32], adding carbonyl units to the BTBT π-system significantly increased the hydrophobicity of the corresponding films. Since both **C₈-BTBT** and **D(C₇CO)-BTBT** have the same hydrophobic BTBT π-system

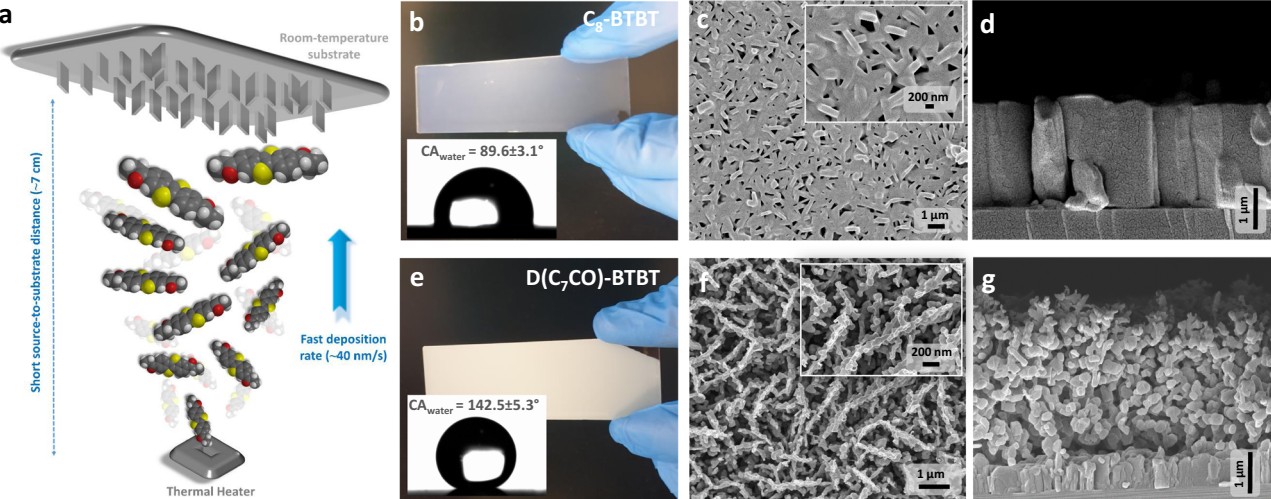

**Fig. 2 Fabrication and characterization of D(C₇CO)-BTBT and C₈-BTBT films. a** Schematic illustration of the physical vapor deposition (PVD) method and the key deposition parameters used for the fabrication of nanostructured SERS films. Photos (**b** and **e**) of the nanostructured films and a water droplet on them, water contact angles (CA$_{water}$), top-view and cross-sectional SEM images of the nanostructured films for **C₈-BTBT** (**c** and **d**) and **D(C₇CO)-BTBT** (**f** and **g**). Scale bars are shown separately for each SEM image.

and medium-length alkyl chains, much larger hydrophobicity (ΔCA$_{water}$ ~50–55°) of the **D(C₇CO)-BTBT** film is attributed to carbonyl-induced (vide infra) growth of three-dimensional highly porous nanostructured morphology on the film surface.

The crystallinity and microstructure of the current films were characterized by conventional θ–2θ diffraction scans which, in combination with the reported single-crystal unit cells and the simulated powder patterns (Supplementary Fig. 4)[23,26], revealed out-of-plane diffraction planes, intermolecular interactions, and the corresponding molecular orientations relative to the substrate surface. As shown in Fig. 3a, **D(C₇CO)-BTBT** film showed five strong high-angle diffraction peaks in the out-of-plane direction corresponding to various crystalline planes (i.e., (111), (511), (020), (021), and (131)), all with nearly face-on oriented molecular π-backbones (Fig. 3b). However, **C₈-BTBT** film showed only two high-angle diffraction peaks (i.e., (020) and (111) planes) with much lower intensities. For both **D(C₇CO)-BTBT** and **C₈-BTBT** films, some degree of edge-on (i.e., (200) and (001), respectively, and higher-order peaks) molecular orientations were evident in the XRDs. The BFDH (Bravais, Friedel, Donnay, and Harker) theoretical crystal morphologies[24,33] predicted plate-like morphologies for both molecules with in-plane crystal growth for edge-on molecular orientations and out-of-plane oriented crystalline domains for face-on oriented molecules (Fig. 3b). In **C₈-BTBT** film, the relatively large intensity of the (001) first-order diffraction peak as compared to (111) and (020) is very consistent with the observed SEM morphology (Fig. 2c), which indicates that **C8-BTBT** molecules favor to remain in the edge-on molecular orientation during the entire PVD process and promote in-plane crystal growth in an island-based morphology. On the other hand, the presence of strong high-angle diffraction peaks (i.e., (111), (511), (020), (021), and (131)) leading to vertically oriented nanoplates (according to BFDH) in **D(C₇CO)-BTBT** film correlates perfectly with the observed SEM morphology (Fig. 2f). The obvious improvement in the morphology of **D(C₇CO)-BTBT** as compared with **C₈-BTBT** originates from the presence of highly polar (μ ~2.9 D) and electron-deficient carbonyl functionalities, which apparently induces very favorable out-of-plane crystal growth. The analysis of the crystalline planes in the **D(C₇CO)-BTBT** film forming the top-lying 3D morphology revealed that this 3D morphological region has a continuous zigzag motif of head-to-tail C=O(δ⁻)⋯(δ⁺)C=O (3.52 Å) (∠

O⋯C=O ~ 95.8° and ∠ C=O⋯C ~ 139.9°) interactions (Fig. 3c). In addition, the acidity of the aliphatic α-methylene hydrogens seems to yield weak (α-*methylene*) C-H⋯O=C (*carbonyl*) (2.87 Å) hydrogen bonds along with the slipped-stacked molecular packing (Fig. 3c)[34–36]. The out-of-plane crystal growth was also found to involve strong (*Ph*) C-H⋯π (*BTBT*)/S⋯π (*thiophene*) (a = 3.27 Å and c = 2.78 Å < r$_{vdw}$(H) + r$_{vdw}$(C) = 2.90 Å/b = 3.22 Å < r$_{vdw}$(S) + r$_{vdw}$(C) = 3.50 Å) and S⋯S (3.43 Å < r$_{vdw}$(S) + r$_{vdw}$(S) = 3.60 Å) interactions (Fig. 3c)[37]. The minimum (*Ar*) C-H⋯π/S⋯S interaction distances were found to be ~0.2 Å smaller than those in **C₈-BTBT**, which is undoubtedly owing to the electronic effect of carbonyl presence (i.e., deshielding on the aromatic π-system and increased acidity for the α-methylene hydrogens)[26]. On the basis of the measured distance, carbonyl⋯carbonyl interactions in **D(C₇CO)-BTBT** film is found to be of dipolar nature[30,32,38], rather than n-π* interactions (typical distances <3.1–3.2 Å)[4,31]. These dipolar head-to-tail carbonyl interactions were reported to be strong noncovalent interactions that could indeed be competitive with hydrogen bonds in some biological molecules[30–32]. On the basis of these findings, the presence of carbonyl units in **D(C₇CO)-BTBT** clearly strengthens intermolecular interactions and facilitates crystal growth in the out-of-plane direction. Indeed, as direct evidence of enhanced cohesive energetics in the solid-state after carbonyl functionalization, one should also note the large melting point increase of ~135 °C in **D(C₇CO)-BTBT** as compared with its non-carbonyl analog **C₈-BTBT**. Our film deposition attempts using unsubstituted **BTBT** (Fig. 1a) also yielded only dense 2D island-based film morphologies with very limited out-of-plane features (Supplementary Fig. 5), which is consistent with the previous reports on thin-films of BTBTs[23].

The cross-sectional SEM images reveal the film growth behaviors of the current molecules. Although **C₈-BTBT** (Fig. 2d) showed consistent growth of densely packed island-based morphology (VW growth mode)[24,39] during the course of the entire film deposition process, a quite unique two-stage film growth behavior was evident for **D(C₇CO)-BTBT** (Fig. 2g). Two distinct growth modes were identified for **D(C₇CO)-BTBT**. Similar to that observed in **C₈-BTBT**, the initial ~600 nm of the **D(C₇CO)-BTBT** film showed dense packing of island-based morphology (VW growth mode), which is obviously a direct result of our modified PVD film growth dynamics[24,29] and the

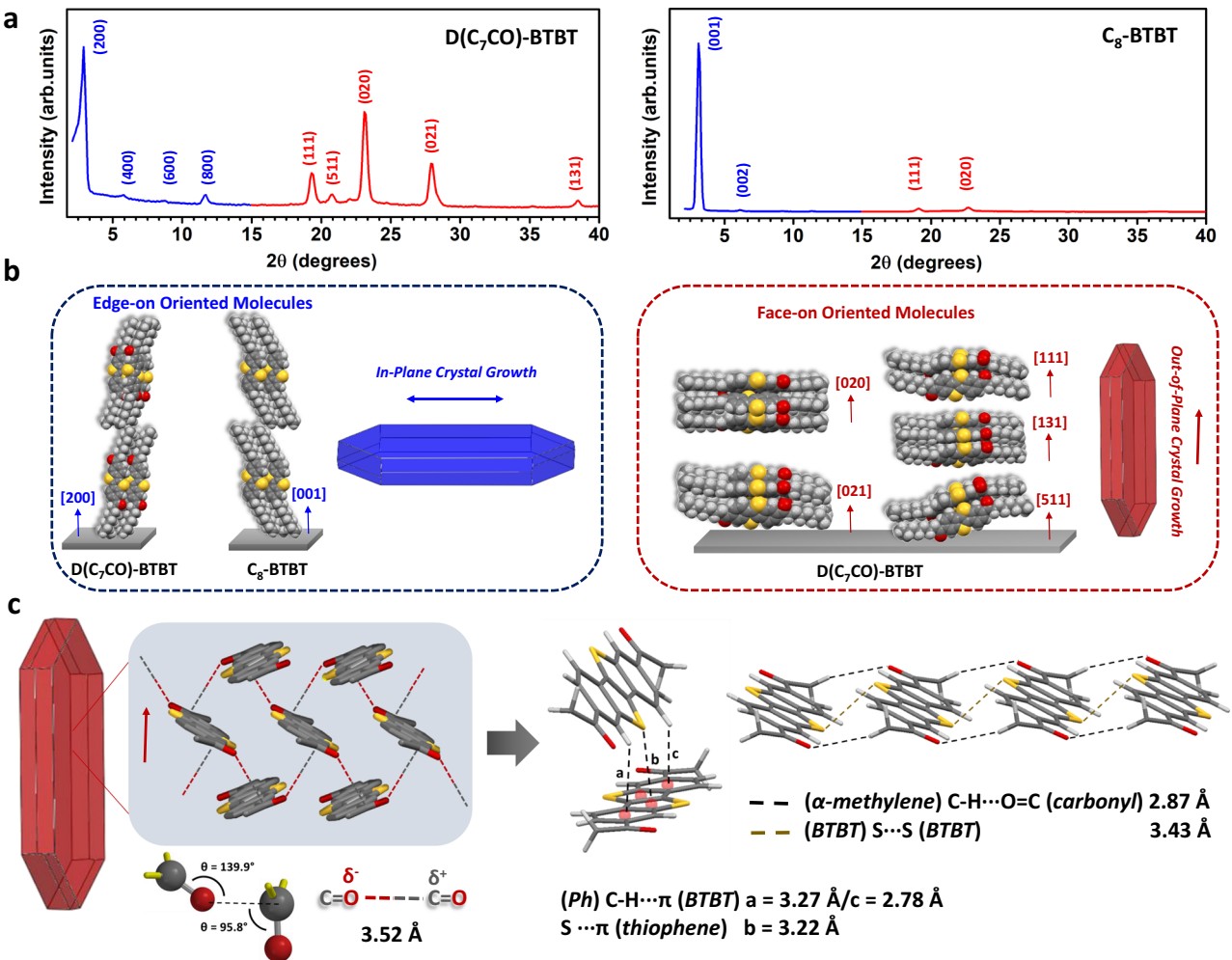

**Fig. 3 XRD characterization and crystal packing in D(C$_7$CO)-BTBT and C$_8$-BTBT films. a** The θ-2θ XRD patterns of the **D(C$_7$CO)-BTBT** and **C$_8$-BTBT** films and the assigned out-of-plane crystallographic planes. **b** The schematic packing illustrations of the diffraction planes identified in the out-of-plane direction and BFDH (Bravais, Friedel, Donnay, and Harker) theoretical crystal growth morphologies formed along with these crystallographic directions. **c** The zigzag structural motif of *head-to-tail* carbonyl···carbonyl interactions in the out-of-plane direction with the corresponding (δ$^-$)O···C(δ$^+$) distance and angles. Perspective views of the herringbone and slipped-stacked molecular packings and key intermolecular interactions in the out-of-plane direction.

tendency of the crystal growth with lowest-energy surfaces ((200) for **D(C$_7$CO)-BTBT** and (001) for **C$_8$-BTBT**-based on BFDH)[40] covering the high-energy substrate surface ($\gamma = 52.5$ mJ/m$^2$ for native oxide)[41]. Once this dense film is formed, growth mode in **D(C$_7$CO)-BTBT** shows the transition into a highly porous 3D morphology forming loosely connected and vertically oriented nanoplates in the top-lying ~4.5–5.0 μm.

In order to understand the origin of this dynamic growth behavior for **D(C$_7$CO)-BTBT**, we first identified the distinct molecular packing in each morphological layer. A controlled short PVD deposition onto Si(001) substrate yielded the first ~600 ± 50 nm layer of densely packed island-based morphology without forming the top-lying 3D morphology (Fig. 4a). The out-of-plane XRD characterization of this film (Fig. 4b) confirmed that this initial layer consists of mainly edge-on oriented molecules that show 2D crystalline growth on the substrate plane as also predicted by the BFDH (Bravais, Friedel, Donnay, and Harker)[24,33] theoretical crystal morphology (shown in blue in Figs. 3 and 4). On the basis of this finding, the higher angle diffractions (2θ = 15–40°) in thick **D(C$_7$CO)-BTBT** films are assigned to the top-lying highly porous 3D morphological layer, which is also consistent with the BFDH theoretical crystal morphology (shown in red in Figs. 3 and 4). The short PVD deposition allowed us to understand that the initial layer of densely

packed island-based morphology gives a highly hydrophobic surface (CA$_{water}$ = 130.4 ± 4.2°, $\gamma = 25.43$ mJ/m$^2$), which further increases to a CA$_{water}$ value of 142.5 ± 5.3° ($\gamma = 22.99$ mJ/m$^2$)) with the contribution of the top-lying 3D porous morphology. Since surface energy and hydrophobicity of the active deposition area are very likely to affect molecular orientation, nucleation/crystal growth, and film morphology[39,42,43], we performed additional depositions on freshly-prepared hydrophobic (CA$_{water}$ ~93.5 ± 2.4° and $\gamma = 41.03$ mJ/m$^2$) polystyrene (PS, $M_n = 5.2$ kDa/PDI = 1.06) grafted SiO$_2$(300 nm)-Si(100) substrates[44,45], and compared these SEM images (Fig. 5) to those fabricated directly on the native oxide silicon substrates (Fig. 2d and g). **D(C$_7$CO)-BTBT** formed porous 3D morphology from the beginning of the film growth when deposited onto the hydrophobic PS-grafted surface. This suggests that the growth of the porous 3D morphology is induced by the surface hydrophobicity onto which molecules are deposited, which could be provided either with a PS-grafted layer or initially deposited island-based **D(C$_7$CO)-BTBT** layer itself. Similar morphology/crystallinity transitions were observed previously for pentacene film deposition (from thin-film phase to bulk phase) after a critical organic layer thickness was reached[43,46]. Also, various *n*- and *p*-type molecular semiconductors showed a 2D-to-3D transition on graphene in their growth modes during film deposition under vacuum[29]. Here, an

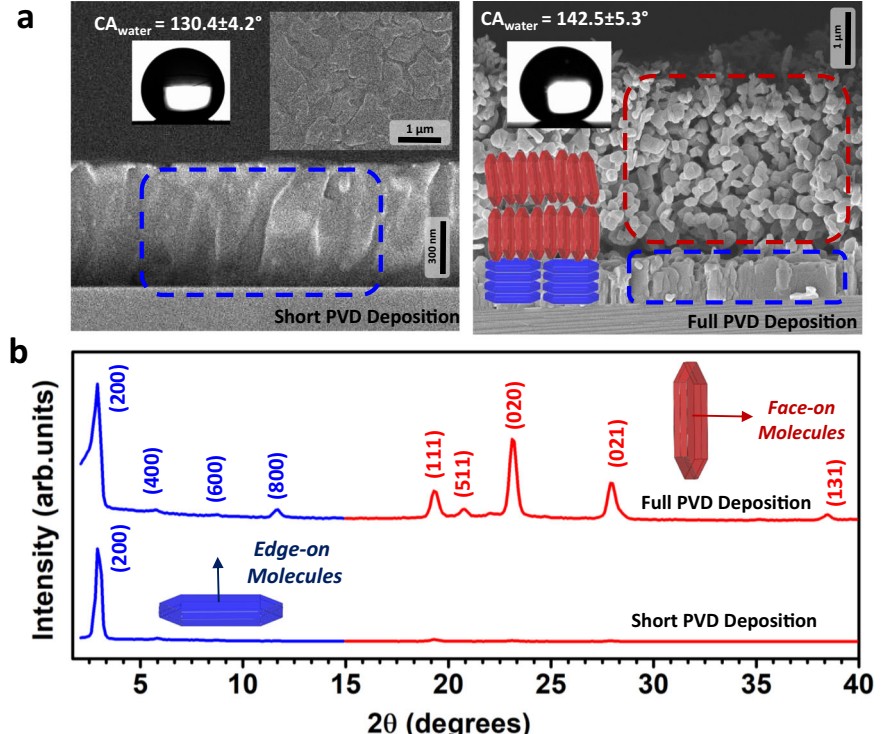

**Fig. 4 Characterization of two-stage film growth in D(C$_7$CO)-BTBT film. a** Cross-sectional and top-view SEM images of the nanostructured films of **D(C$_7$CO)-BTBT** for short (~600 ± 50 nm thick film) and full (~5.5 ± 0.2 µm thick film) deposition conditions, photos of the water droplets on them with the corresponding contact angles (CA$_{water}$). The corresponding edge-on and face-on-oriented molecular domains are shown in blue and red, respectively. Scale bars are shown separately for each SEM image. **b** The θ-2θ XRD patterns of the nanostructured films of **D(C$_7$CO)-BTBT** for short (~600 ± 50 nm thick film) and full (~5.5 ± 0.2 µm thick film) deposition conditions along with the assigned out-of-plane crystallographic planes, BFDH (Bravais, Friedel, Donnay, and Harker) theoretical crystal morphologies and the relative directions of "edge-on" and "face-on" oriented molecular packings. Note that herein the terms "edge-on (shown in dark blue)" and "face-on (shown in red)" refer to the molecular orientations with respect to the substrate plane.

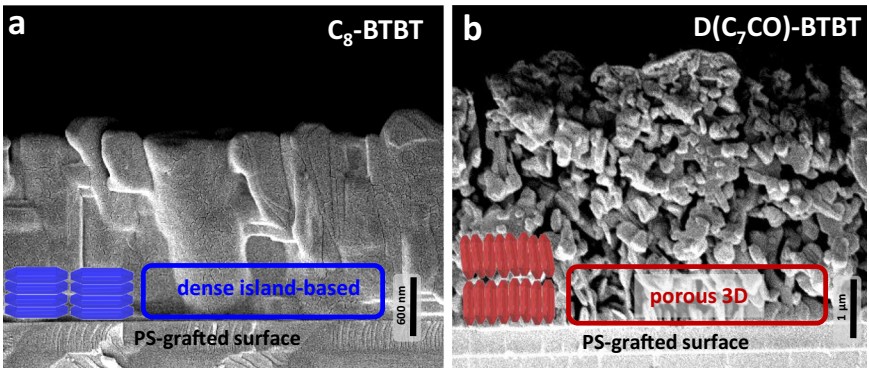

**Fig. 5 SEM characterization of C$_8$-BTBT *vs.* D(C$_7$CO)-BTBT films on PS-grafted surfaces.** Cross-sectional SEM images of the nanostructured films of **C$_8$-BTBT** (**a**) and **D(C$_7$CO)-BTBT** (**b**) on hydrophobic PS-grafted surfaces. The corresponding dense island-based (in blue) and porous 3D (in red) morphologies, and the relative directions of "face-on" (in blue) and "edge-on" (in red) oriented BFDH (Bravais, Friedel, Donnay, and Harker) theoretical crystal morphologies were shown for the initial stages of the organic film growths. Scale bars are shown separately for each SEM image.

interesting question arises as to why such a thick (~600 nm) **D(C$_7$CO)-BTBT** layer is needed to induce 3D morphological growth. We note that this layer was indeed formed in only 15 s based on the ultra-high film growth rate (~40 nm/s) of our PVD process. Therefore, this densely packed initial layer was formed in a relatively short time before the deposition process reaches a thermodynamic equilibrium to induce the continued 3D growth. We believe that this initial layer could be made much thinner with a more-controlled deposition rate, which seems not to be important for our study at this point since the top ~1–2 µm morphology is crucial for SERS activity. On the other hand, the PS-grafted hydrophobic

surface still resulted in dense island-based film growth for **C$_8$-BTBT** with no porous 3D morphology. Therefore, it seems that there are threshold surface properties (i.e., surface energy and hydrophobicity), probably unique to each molecular structure, essential to induce 3D porous morphological growth. A crystal growth on a substrate that does not meet these properties would induce only 2D crystal growth since it requires the lowest-energy crystalline surface, that is the largest area surface in the BFDH morphology[40], to be on the substrate plane. The combination of intrinsic factors such as intermolecular (dipolar carbonyl, strong C-H···π/S···π/S···S, and weak C-H···C = O hydrogen bonding) interactions along the crystal growth

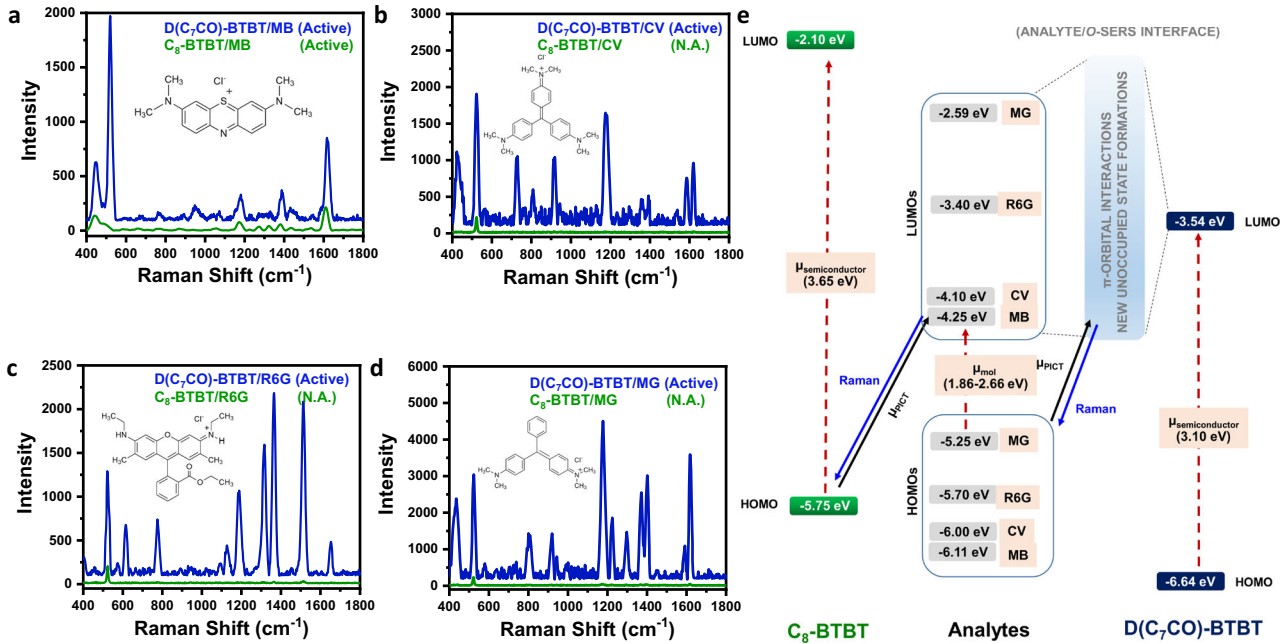

**Fig. 6 Raman enhancement for D(C₇CO)-BTBT and C₈-BTBT films, and chemical enhancement mechanism.** SERS spectra of methylene blue (MB) (**a**), crystal violet (CV) (**b**), rhodamine 6 G (R6G) (**c**), and malachite green (MG) (**d**) analytes on **D(C₇CO)-BTBT** and **C₈-BTBT** films. Insets depict the molecular structures of the analyte molecules. **e** Energy level diagram for the current analyte/o-SERS systems for **D(C₇CO)-BTBT** and **C₈-BTBT** showing the frontier molecular orbital (HOMO/LUMO) energies and plausible (shown with solid black arrow)/non-plausible (shown with dashed red arrows) transitions. $\mu_{PICT}$ stands for photoinduced charge-transfer between analyte and semiconductor under the excitation of 785 nm. Raman signals were produced during the transitions shown with blue solid arrows.

direction and extrinsic factors such as surface energy and hydrophobicity onto which molecules are deposited were found to be the key factors in the formation of porous 3D morphology for **D(C₇CO)-BTBT** film with vertically oriented nanoplates. The top morphology of the **D(C₇CO)-BTBT** film is highly favorable for SERS detection since it enables a large surface area for the analyte deposition and allows for effective π-interactions between the analyte and face-on **D(C₇CO)-BTBT** molecules in varied crystalline orientations[17,18].

**Raman signal enhancements for nanostructured D(C₇CO)-BTBT and C₈-BTBT films, and the origins of the chemical enhancement mechanism.** The effects of BTBT molecular design, carbonyl functionalization, and electronic structure on intermolecular interactions and film growth (microstructure/morphology) mechanisms have been discussed in detail in the previous section. In this section, the Raman signal enhancement abilities of the current 3D-nanostructured **D(C₇CO)-BTBT** and 2D-morphology dominant **C₈-BTBT** films were investigated using four different analyte molecules (i.e., MB[47], rhodamine 6 G (R6G)[48], crystal violet (CV)[49], and malachite green (MG)[50]) with varied electronic structures and HOMO/LUMO levels spanning 1.0 and 1.5 eV energy ranges, respectively (Fig. 6). These molecules were drop-casted from their aqueous solutions ($1 \times 10^{-3}$ M) onto semiconductor films, and a Raman laser excitation wavelength of 785 nm (1.58 eV) was employed in the SERS experiments, at which both semiconductors and the analyte molecules exhibit no electronic transitions ($\lambda_{abs}^{\text{low-energy onset}}$'s < 700 nm, Supplementary Fig. 6). Therefore, purely analyte electronic excitation ($\mu_{mol}$) or semiconductor exciton (band gap) resonances ($\mu_{semiconductor}$) could not contribute to the Raman enhancement in the current analyte/o-SERS systems.

As shown in Fig. 6 and Supplementary Fig. 7, whereas 2D-morphology dominant **C₈-BTBT** films did not show any Raman signal enhancement with R6G, CV, and MG analyte molecules and

showed SERS signals only for MB (green spectrum in Fig. 6a) with low intensities (I's < 250), nanostructured **D(C₇CO)-BTBT** films showed substantial Raman enhancements for all analyte molecules with I's of up to 4000–5000. The Raman shifts and the assignments for the observed bands are listed in Supplementary Table 1. The prominent Raman peaks for MB ($\nu$(C–C) ring stretches at 1621 cm⁻¹, $\delta$(C-N-C) skeletal deformation at 445 cm⁻¹, and $\nu$(C-N) symmetric/asymmetric stretches at 1391/1435 cm⁻¹) are slightly different than those observed on metallic platforms[17,24], indicating the presence of analyte-semiconductor interaction specific signal enhancements[51]. We note that, despite the low performance on **C₈-BTBT** films, a molecular-specific (i.e., among a sample of four analyte molecules) Raman enhancement is achieved on an organic-SERS platform[52]. Considering that undoped π-conjugated organic semiconductors have a low intrinsic free carrier density ($10^{13}$ carrier/cm³)[17], which is much lower than those of metals ($10^{22}$–$10^{23}$ carrier/cm³)[53], electromagnetic contribution to SERS is an implausible mechanism for the current organic films. Therefore, as demonstrated in the previous reports by Lombardi et al.[51,54,55] and our research groups[17,18] on (in)organic-SERS platforms, CT resonance occurred at the analyte/semiconductor interface should be the key player to enhance the polarizability derivative tensor for analyte vibrational modes, which consequently leads to Raman signal enhancements. We also note that the CT resonances could couple with 3D morphology-driven Mie scattering to further benefit from the porous nanostructured morphology in **D(C₇CO)-BTBT** films[56]. The formation of CT resonances requires strong π-orbital interactions (mixing) between analyte and semiconductor molecules at the interface. When the semiconductor molecules have a properly oriented π-orbital system for effective spatial overlap, which is a face-on orientation towards the analyte-semiconductor interface, and there is a small energy difference between the corresponding (i.e., analyte and semiconductor) wave functions, a strong interfacial state formation could be expected. Our previous studies

indeed revealed that π-electron-deficient (π-delocalized LUMOs with $E_{LUMO} < -3$ eV) **DFH-4T** and **DFP-4T** molecules, in their properly oriented nanostructured films, formed resonant CT-states with MB and R6G analytes[17,18]. As evidenced by the microstructural analysis for **C$_8$-BTBT** (vide supra), the edge-on π-backbone dominant orientation with σ-insulating alkyl substituents pointing towards the analyte-semiconductor interface precludes an efficient π-π mixing in the case of analyte/**C$_8$-BTBT** pair for the majority of this semiconductor film. Therefore, on **C$_8$-BTBT** film surface, there is only a limited region with face-on semiconductor π-backbones, which in the first place deteriorates Raman enhancements (Г's < 250). Second, the only Raman enhancement observed on **C$_8$-BTBT** was with the lowest-LUMO (−4.25 eV) analyte molecule, MB, in which an energetically feasible resonant transition (~1.50 eV based on vacuum energy levels) could be defined from semiconductor to analyte frontier π-orbitals ($\psi_{HOMO(C8-BTBT)} \rightarrow \psi_{LUMO(MB)}$, μ$_{PICT}$ in Fig. 6e). However, when it comes to the nanostructured **D(C$_7$CO)-BTBT** film, energetically feasible transitions in resonance with the incident photon energy (1.58 eV (785 nm)) could not be identified between the pristine analyte and **D(C$_7$CO)-BTBT** frontier π-orbitals since **D(C$_7$CO)-BTBT**'s HOMO is stabilized by ~0.9 eV. As a result, all plausible transitions would require energies larger than 2 eV. Despite this energetic mismatch, the fact that all four analyte molecules showed strong Raman enhancements suggests that there should be strong frontier π-orbital interaction at the semiconductor-analyte interface on **D(C$_7$CO)-BTBT** film-forming new interfacial states. In order to understand the origin of these frontier π-orbital interactions, the energy level diagram for both semiconductors and the analyte molecules is depicted in Fig. 6e. The first point to be made from this diagram is that going from **C$_8$-BTBT** to **D(C$_7$CO)-BTBT**, the semiconductor's HOMO is stabilized by ~0.9 eV that results in an energy mismatch between the HOMOs. If the π-orbital interactions at the analyte/semiconductor interface were to occur between the HOMOs, from an energetic standpoint, one would expect an exact opposite SERS finding that **C$_8$-BTBT** film shows SERS activity with all analyte molecules and **D(C$_7$CO)-BTBT** is SERS-inactive with some of these analytes. This, however, is not the case in our SERS measurements. Therefore, based on our SERS results, the key interfacial π-orbital mixing should be between the LUMO's of the semiconductor and analyte molecules. From a purely quantum mechanical perspective, the higher energy and larger spatial extension of LUMO levels, when compared to HOMOs, make them more likely to be perturbed and interact in complex π-systems, especially for intermolecular orbital interactions that require a spatial overlap of the corresponding wave functions[57]. Consistent with the discussions made earlier for **D(C$_7$CO)-BTBT** vs. **C$_8$-BTBT**, whereas the LUMO energy level of **C$_8$-BTBT** is much higher (up to ~2 eV energy mismatch) than those of analyte molecules, **D(C$_7$CO)-BTBT**'s LUMO is more π-extended and stabilized due to carbonyls, and becomes energetically close to those of the analytes. As shown in Fig. 6e, **D(C$_7$CO)-BTBT**'s LUMO lies symmetrically with respect to those of all analyte molecules. This enables significant wave function mixing with the π-conjugated analyte molecules to yield new interfacial unoccupied states for low-energy CT transitions. The Raman enhancements observed in the nanostructured **D(C$_7$CO)-BTBT** films most likely benefit from photoinduced "$\psi_{HOMO(analyte)} \rightarrow \psi_{new\ unoccupied\ states}$" charge transfers (μ$_{PICT}$ in Fig. 6e). It is noteworthy that since **D(C$_7$CO)-BTBT** film surface includes varied crystalline planes (i.e., (111), (511), (020), (021), and (131)) with all nearly face-on oriented molecules, semiconductor-analyte π-interactions could occur in various geometries to find the most favorable orientation for each vibrational mode.

As previously demonstrated by Linic and co-workers, CT processes within a complex metal-adsorbate system can be identified experimentally via SERS[58]. This identification is mainly based on the elevation of anti-Stokes Raman signal intensities, which is the direct result of increased population of excited vibrational states compared to the equilibrated states when charge excitation occurs on a π-system[59,60]. Therefore, for a defined vibrational mode, the ratio of anti-Stokes to Stokes signal intensities indicates CT efficiency. To make a direct comparison, MB analyte was employed since it is the only molecule to exhibit SERS activity on both semiconductor surfaces. The Stokes and anti-Stokes SERS spectra were collected under the laser excitation of 1.58 eV. As shown in Fig. 7, we first note that the anti-Stokes signal intensities collected on **D(C$_7$CO)-BTBT** film are much higher than those on **C$_8$-BTBT** film. To better understand these observations, the ratios of anti-Stokes/Stokes Raman signal intensities ($\rho_{SERS}$) for specific MB vibrational modes ($\nu_m$) were calculated and compared with the ratio for a similar energy mode in liquid toluene ($\rho_{tol}$) as demonstrated earlier[61,62]. The calculations were carried out using the equation shown in Fig. 7a, in which $K$ describes the degree by which the SERS anti-Stokes signal exceeded the expectation of the Boltzmann distribution. Although $I_{aS}^{SERS}$ and $I_S^{SERS}$ are the observed anti-Stokes and Stokes signal intensities, respectively, for the analyte molecule on the current films, $I_{aS}^{tol}$ and $I_S^{tol}$ are the anti-Stokes and Stokes signal intensities, respectively, for toluene. The calculated $K$ values for **D(C$_7$CO)-BTBT/MB** were found to be large and in the range of 5.8–22.3, whereas those of **C$_8$-BTBT/MB** were much smaller (2.1–6.5). These results provide further evidence that interfacial CT processes are more effective within the **D(C$_7$CO)-BTBT/MB** system when compared with the **C$_8$-BTBT/MB** system.

## Discussion

Highly crystalline nanostructured porous organic films comprised of π-electron-deficient (low-LUMO) thienoacene molecule **D(C$_7$CO)-BTBT** have been fabricated via PVD and showed great promise in organic-SERS applications. Following our recent breakthroughs with oligothiophenes, a completely different π-framework and a unique functionalization strategy are demonstrated. Different than our previous SERS-active oligothiophenes, the current semiconductor was prepared on the gram-scale in ambient via facile Friedel-Crafts acylation reaction without any high-cost transition-metal catalyst. The purification was also based on simple precipitation/solvent washing (i.e., no need for tedious chromatography/sublimation). On the basis of straight comparison to the non-functionalized analog, **C$_8$-BTBT**, at the molecular level and detailed film characterizations, film growth mechanisms have been revealed, and polar carbonyl functionalization was found to enable proper nanostructured film formation (out-of-plane crystal growth) with face-on π-backbones via dipolar C=O⋯C=O interactions, hydrogen bonds, and strengthened π-interactions. The analysis of the electronic structures and the ratio of the anti-Stokes to Stokes SERS suggest that the π-extended and stabilized LUMOs with crystalline face-on orientations in varied directions, all of which are the direct results of carbonyl functionalization, are key for the chemical enhancement mechanism to be effective. Instead of running more complicated electronic structure calculations involving analyte/o-SERS complexes and excited-state characteristics, as we have previously performed[17,18], our analysis based on frontier orbitals vacuum energy levels herein provides facile guidance to elucidate SERS ability of π-conjugated semiconductors. Using **C$_8$-BTBT** platform with a relatively high-lying LUMO that does not allow for effective π-mixing with the corresponding orbitals of the analytes, molecular sensitivity for Raman enhancement is also achieved on an organic-SERS platform. Employing a

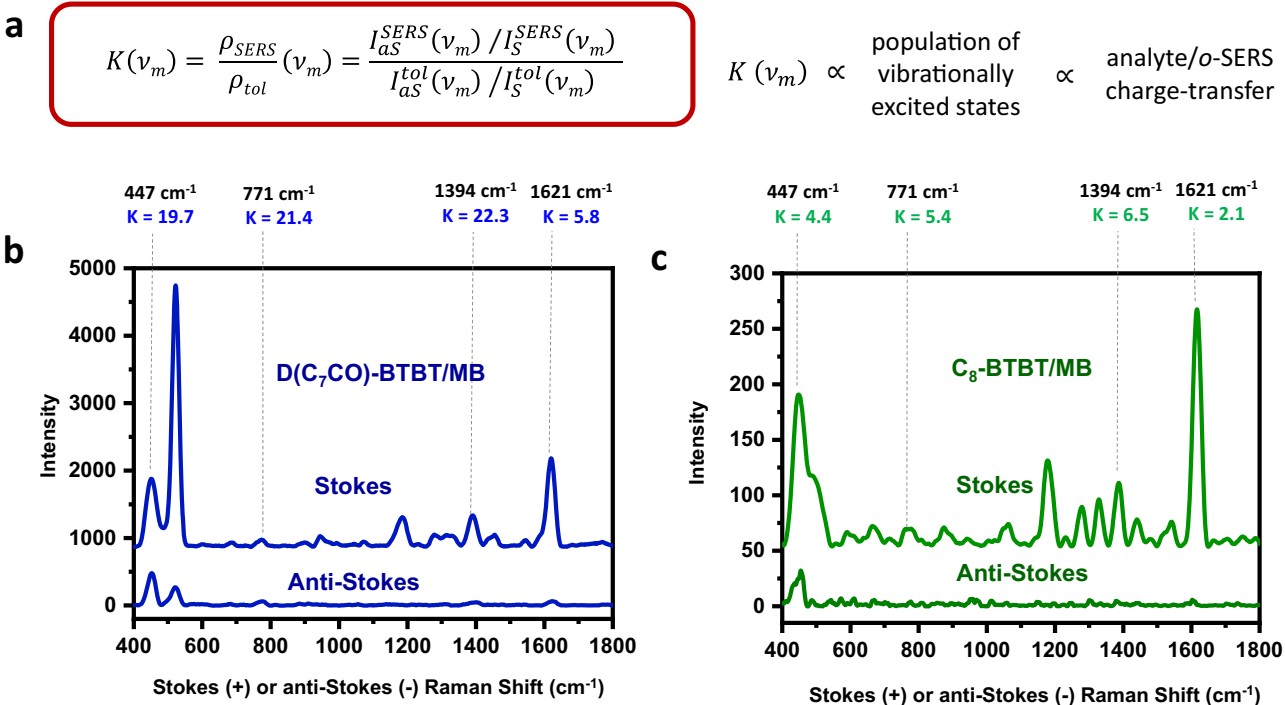

**Fig. 7 Stokes vs. anti-Stokes Raman spectra for D(C₇CO)-BTBT and C₈-BTBT films. a** The equation describing the degree ($K$) by which the SERS anti-Stokes signal exceeded the expectation of the Boltzmann distribution. Stokes and anti-Stokes Raman spectra for **D(C₇CO)-BTBT/MB** (**b**) and **C₈-BTBT/MB** (**c**) analyte/*o*-SERS systems, and the calculated $K$ values for selected Raman peaks ($\nu_m$). $I_{aS}^{SERS}$ and $I_S^{SERS}$ are the observed anti-Stokes and Stokes signal intensities, respectively, for the analyte molecule on the current films. $I_{aS}^{tol}$ and $I_S^{tol}$ are the anti-Stokes and Stokes signal intensities, respectively, for toluene.

future research direction towards carbonyl-functionalized (hetero) acenes having relatively easy synthesis-purification could bring in significant opportunities for organic-SERS platforms such as low materials cost, gram-scale synthesis, structural versatility for potential SERS-based molecular sensors, and solution-based film fabrication. These are envisioned based on the following facts: (i) Friedel-Crafts acylation is a highly universal reaction that could be performed on most of the π-systems and does not require a pre-existing group (e.g., triaalkylstannyl, boronic esters/acids, or halides in cross-coupling reactions), which gives great flexibility to employ various (hetero)acenes, (ii) single or multiple ("di-" in this study) functionalization(s) could be performed and the substituent(s) adjacent to the carbonyl unit could indeed be of any chemical structure, which gives a great tuning ability for molecular/electronic structures, and (iii) proper substituents adjacent to the carbonyl unit would increase molecular solubility, which could allow for the fabrication of solution-processed semiconductor thin-films that could be exposed to a post-deposition process (e.g., thermal or solvent annealing) to form nanostructured surfaces. Especially, mono-carbonyl functionalization might give a proper tuning of LUMO (i.e., not as low as dicarbonyl derivatives) for enhanced molecular sensitivity. As another potential future direction from a molecular engineering perspective, different polar EWD-functionalities (e.g., cyano-, dicyanovinylene-, or nitro-) that provides frontier orbital tuning and π-delocalization (negative meso-meric effect) at the same time could also be employed to explore similar nanostructured films and Raman enhancements. Future studies with other polar EWD unit-functionalized (hetero)acenes and additional analyte molecules would provide a better under-standing of the role of (hetero)acenes, polar EWD units, and substituents in the formation of nanostructured highly crystalline semiconductor films and the role of LUMOs in determining SERS activity and molecular sensitivity.

## Methods

**Synthesis, chemical characterization, computational/crystal analysis, and fabrication/characterization of organic semiconductor films.** BTBT-based π-conjugated semiconductors 1,1'-(benzo[b]benzo[4,5]thieno[2,3-d]thiophene-2,7-diyl) bis(octan-1-one) (**D(C₇CO)-BTBT**) and 2,7-dioctyl[1]benzothieno[3,2-*b*][1]ben-zothiophene (**C₈-BTBT**) were used as molecular building blocks in physical vapor deposition. Prior to the semiconductor film fabrication, Si wafers (001 crystallographic orientation and 1–10 Ω resistivity) were cleaned in the sonicating bath with acetone and ethanol (10 min each) and dried with nitrogen gas. The wafers were then treated with piranha solution for 1 h, washed with DI water, and dried with nitrogen gas. To further remove contaminants from the wafer surface, pre-cleaned samples were treated with UV-ozone cleaner for 15 min. For micro-/nanostructured semiconductor film depositions, **D(C₇CO)-BTBT** or **C₈-BTBT** powder (10-20 mg) was placed in a tungsten boat and thermally evaporated using a conventional PVD system (NANO-VAK HV) under high vacuum ($1 \pm 0.2 \times 10^{-6}$ Torr). During film deposition, a 90° deposition angle along with an ultrafast deposition rate (~40 nm s⁻¹) and a short source-to-substrate distance (~7 cm) were employed. The morphology and micro-structure of the deposited films were characterized with a scanning electron micro-scope (Zeiss, Gemini-SEM 500 Field Emission SEM) and X-ray diffraction (Malvern Panalytical, Empyrean X-Ray Diffractometer). UV-vis absorption and water contact-angle measurements were carried out using a Shimadzu 2600 UV-vis-near-IR spec-trophotometer and a Krüss, DSA 100 drop shape analyzer, respectively. PS-SiO₂(300 nm)-Si(100) substrates were prepared by using grafting-to method (ω-hydroxy-terminated poly(styrene) from Polymer Source Inc., Canada, $M_n$ = 5.2 kDa and PDI = 1.06) onto SiO₂(300 nm)-Si(100) substrates and in accordance with the reported procedures[44,45]. The theoretical morphology for **D(C₇CO)-BTBT** was cal-culated according to the BFDH (Bravais, Friedel, Donnay, and Harker) method using the program Mercury 2.4 (CCDC).

**SERS experiments.** Raman signal enhancement behavior of the films was inves-tigated using MB, R6G, CV, and MG as reporter molecules at an excitation wavelength of 785 nm. In a typical experiment, 5.0 μL aqueous analyte solution with a concentration of $10^{-3}$ M was dripped onto the fabricated films followed by storing in a hood at room temperature until dry. Raman spectra were subsequently collected from at least ten different spots across the entire dried area using a high-resolution confocal Raman spectrometer (Jasco NRS-4500). A ×20 objective lens with a laser spot diameter of ~3 μm, and 30 mW laser power was used to obtain all Raman spectra. Acquisition time was also 10 s for SERS investigation of the films and 2 s for Stokes and anti-Stokes Raman investigation.

**Synthesis and structural characterization.** The reactions in ambient were carried out by simply capping the reaction flask with a rubber septum. Conventional Schlenk techniques were used for the reactions performed under $N_2$ atmosphere. All chemicals were purchased from commercial sources and used without further purification unless otherwise noted. $^1H/^{13}C$ NMR spectra were recorded on a Bruker 400 spectrometer ($^1H$, 400 MHz; $^{13}C$, 100 MHz). Elemental analyses were done on a LecoTruspec Micro model instrument. High-resolution mass spectra were measured on a Bruker Microflex LT MALDI-TOF-MS Instrument. The optimization of the molecular geometries and the analysis of the frontier molecular orbitals were carried out by Gaussian 09 using density functional theory (DFT) at B3LYP/6-31 G** level[63]. The synthesis of [1]benzothieno[3,2-b][1]benzothiophene (**BTBT**) and 2,7-dioctyl[1]benzothieno[3,2-b][1]benzothiophene (**$C_8$-BTBT**) were carried out in accordance with the reported procedures[23,28].

Synthesis of 1,1'-(benzo[b]benzo[4,5]thieno[2,3-d]thiophene-2,7-diyl) bis(octan-1-one) (**D($C_7$CO)-BTBT**): $AlCl_3$ (3.03 g, 22.76 mmol) was added into a solution of [1]benzothieno[3,2-b][1]benzothiophene (1.5 g, 6.24 mmol) in anhydrous dichloromethane (150 mL) at −10 °C under nitrogen or ambient. The resulting solution was stirred at −10 °C for 30 min. Then, the reaction mixture cooled down to −78 °C. Octanoyl chloride (5.075 g, 31.20 mmol) was subsequently added dropwise, and the mixture was stirred for 1 h at the same temperature. The reaction mixture was then allowed to warm to room temperature and stirred for 2 days under nitrogen or ambient. The reaction mixture was quenched with water and the precipitated solid was collected by vacuum filtration, and washed with water and methanol, respectively. In order to remove mono-acylated by-product from the precipitated solid, a small amount (~100 ml) of dichloromethane was used during solvent washing. The product was obtained as a pale yellow solid (2.09 g, 68% yield-under nitrogen), which showed sufficient purity and was directly used in physical vapor deposition. The reaction yield was ~60% when the reaction was carried out in ambient conditions. Melting point: 265-266 °C; $^1H$ NMR (400 MHz, $CDCl_3$), δ (ppm): 8.59 (s, 2H), 8.09 (d, 2H, $J$ = 8.0 Hz), 7.98 (d, 2H, $J$ = 8.0 Hz), 3.10 (t, 4H, $J$ = 12.0 Hz), 1.78–1.83 (m, 4H), 1.33–1.44 (m, 16H), 0.89–0.92 (t, 6H, $J$ = 16.0 Hz); $^{13}C$ NMR (100 MHz, $CDCl_3$), δ (ppm): 199.5, 142.8, 136.2, 135.8, 134.3, 124.9, 124.6, 121.9, 38.9, 31.7, 29.4, 29.2, 24.5, 22.6, 14.1; MS (MALDI-TOF) $m/z$ calcd for $C_{30}H_{36}O_2S_2$: 492.22 [M$^+$]; found: 492.44 [M$^+$]; elemental analysis calcd (%) for $C_{30}H_{36}O_2S_2$: C, 73.13; H, 7.36; found: C, 73.46; H, 7.67.

## Data availability

The data that support the findings of this study are presented in the manuscript and supplementary information file. Source data are available from the corresponding author upon request. The X-ray crystallographic coordinates for D($C_7$CO)-BTBT and $C_8$-BTBT have previously been deposited at the Cambridge Crystallographic Data Centre (CCDC) under the deposition numbers 1946322 and 679293, respectively. These data can be obtained free of charge from The Cambridge Crystallographic Data Centre via www.ccdc.cam.ac.uk/data_request/cif.

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

## Acknowledgements

H.U., G.D. and I.D. acknowledge support from the Scientific and Technological Research Council of Turkey (TUBITAK) grant number 216M430. G.D. also thanks TUBITAK (grant no: 119C025) for the financial support. We thank professor Fahri Alkan for fruitful discussions on the theoretical perspective of this study.

## Author contributions

H.U. and G.D. conceived and designed the experiments. I.D. and H.U. synthesized the small molecular semiconductors and performed the theoretical calculations. G.L. and G.D. fabricated the organic semiconductor films and performed the SERS experiments. A.C. performed the semiconductor film characterizations. H.U. and G.D. co-wrote the paper. All authors discuss the results and commented on the manuscript.

## Competing interests

The authors declare no competing interests.
