## [Peer Review File · Nature Communications]

REVIEWERS' COMMENTS

Reviewer #1 (Remarks to the Author):

In the manuscript entitled “Enabling Three-Dimensional Porous Architectures via Carbonyl Functionalization: Towards Molecular Specific Organic-SERS Platforms”, the authors Deneme et al. fabricated highly crystalline nanostructured porous organic films comprised of D(C7CO)-BTBT, which is, π -electron deficient (low-LUMO) thienoacene molecule fabricated by adding carbonyl units to BTBT π -system, with low cost, simple synthesis and purification process. Polar carbonyl interactions of D(C7CO)-BTBT and crystal growth into out-of-plane direction leads to highly porous 3D morphology of the fabricated film. The electron-withdrawing (EWD) carbonyl functionalization stabilized and extended π -orbitals and so enhanced SERS signal compared to C8-BTBT, showing great promise in organic-SERS applications. Therefore, I suggest the publication of this manuscript in Nature Communications after addressing minor issues which are,

1. In line 138 at page 5 and line 355 at page 13, the authors claimed that porous surface of D(C7CO)-BTBT film is driven by hydrophobicity. However, in line 217 at page 7, they claimed that larger hydrophobicity of D(C7CO)-BTBT film is attributed to highly porous morphology. The authors should take one side.
2. For the logics above mentioned, the authors should provide explanations about how hydrophobicity increases porosity or vice versa.

Reviewer #2 (Remarks to the Author):

Title: Enabling Three-Dimensional Porous Architectures via Carbonyl Functionalization: Towards Molecular Specific Organic-SERS Platforms

Authors: İbrahim Deneme, Görkem Liman, Ayse Can, Gökhan Demirel, Hakan Usta

The science of surface-enhanced Raman spectroscopy (SERS) has expanded greatly since its inception. Initially all work was done on Ag or Au nanostructured substrates, but these substrates were often volatile or unstable in many ways, and the search for stable and reproducible substrates turned to semiconductors. Semiconductors are not as sensitive as metals as SERS substrates, but a trade-off can be made in order to take advantage of the wide variety of semiconductors available. We are currently in a phase of SERS research in which many of the available semiconductors for SERS substrates are being explored. This variety enables us to design substrates which are specific for a single or small related group of molecules. Metals often lack this specificity which make it difficult to distinguish a single species from a mixture. Enhancements for specific molecules take advantage of the charge-transfer contributions to semiconductor SERS to enable high specificity. There are other advantages of semiconductors, for example some applications require reproducibility, and/or reusability. Once again, metals fail in this, while fabrications of certain semiconductors display such properties. These properties are very important for many practical applications of SERS. We should also note that in practical applications, requiring large-scale numbers of samples for detection, costs are also of importance. Once again, semiconductors are superior to metals in this aspect. This explains why the current work by Deneme et al. is of considerable interest. The authors point the way to produce stable, reproducible and controllable SERS substrates. They take advantage of the underlying theory of SERS to design what I

believe will be a useful approach to fulfilling the promise of SERS.

The authors construct a substrate from nanostructured organic films of π -electron deficient fluorinated oligothiophene semiconductors using physical vapor deposition. It is clear from early work in SERS that large surface area (a 2D property) is of importance to achieve high sensitivity. But 3D structures with high porosity are needed to achieve high surface coverage within the confines of a laser beam. The authors utilize a series of four dye molecules of similar structure (namely MB, R6G, CV and MG) which enable some "scanning" of the HOMOs and LUMOs to allow investigation of the charge-transfer contributions to the Raman intensity, as well as a test of molecular specificity. They also examine the ratio of Stokes to anti-Stokes lines in the spectrum to elucidate the charge-transfer properties as well as the effect on low-lying spectral lines.

This article definitely should be published for its ground-breaking utilization of novel properties of their chosen substrates. If this technique can be applied more broadly, it can have a large effect on the practical applications of SERS. The promise of a practical application of SERS will be a lot closer to being fulfilled.

REVIEWER 1

The reviewer mentions that the electron-withdrawing (EWD) carbonyl functionalization stabilized and extended π -orbitals and so enhanced SERS signal compared to C8-BTBT, showing great promise in organic-SERS applications. He/She suggests the publication of our manuscript in *Nature Communications* after addressing the following minor points. We deeply thank the reviewer for his/her favourable comments. Our responses to his/her comments are as follows:

Reviewer Comment 1. *In line 138 at page 5 and line 355 at page 13, the authors claimed that porous surface of D(C7CO)-BTBT film is driven by hydrophobicity. However, in line 217 at page 7, they claimed that larger hydrophobicity of D(C7CO)-BTBT film is attributed to highly porous morphology. The authors should take one side.*

Reviewer Comment 2. *For the logics above mentioned, the authors should provide explanations about how hydrophobicity increases porosity or vice versa.*

Response to Reviewer Comments 1 and 2. Since these two comments are related, a combined response is provided. We thank the reviewer for his/her helpful comment to make us clarify this point. In D(C7CO)-BTBT film, on the basis of our study, two distinct growth modes were identified. Similar to that observed in C8-BTBT, the initial ~600 nm of the D(C7CO)-BTBT film showed dense packing of island-based morphology (VW growth mode), which is obviously a direct result of our modified PVD film growth dynamics and the tendency of the crystal growth

with lowest energy surfaces ((200) for D(C7CO)-BTBT and (001) for C8-BTBT based on BFDH) covering the high-energy substrate surface ($\gamma = 52.5 \text{ mJ/m}^2$ for native oxide). Once this dense film is formed, growth mode in D(C7CO)-BTBT shows transition into a highly porous 3D morphology forming loosely connected and vertically oriented nanoplates in the top-lying $\sim 4.5\text{-}5.0 \text{ }\mu\text{m}$. Therefore, the surface hydrophobicity of the initially formed island-based layer promotes further hydrophobicity by facilitating the formation of 3D porous morphology. The contact angle increase from 130° to 143° confirms this. However, the final impressive hydrophobicity of the D(C7CO)-BTBT, which is almost superhydrophobicity, is governed by highly porous 3D morphology in the top-lying $\sim 4.5\text{-}5.0 \text{ }\mu\text{m}$, which is indeed facilitated by the initially formed surface hydrophobicity in the first $\sim 600 \text{ nm}$ layer. The cross-sectional SEM images, combined with XRD characterizations and BFDH theoretical modelling, proved this point. Finally, the reason why we see a molecular orientation and crystal morphology transition during film growth is because the lowest energy surface for D(C7CO)-BTBT ((200) based on BFDH modelling), which has the largest surface area, does not need to fully cover the surface anymore, in other words, crystallites could adopt edge-on orientations, when the surface energy decreases by the initially formed island-based layer. Then, it transitions into varied vertical orientations to form the final 3D porous morphology. Therefore, the hydrophobicity of the initially formed island-based layer induces 3D porous morphology formation based on this aforementioned reason. All these points are clearly explained in the manuscript; however, we have made small changes in some sentences to further clarify these points.

REVIEWER 2

The reviewer mentions that our study definitely should be published in *Nature Communications* for the ground-breaking utilization of novel properties for the chosen substrates. He/She adds that if this technique can be applied more broadly, it can have a large effect on the practical applications of SERS, and the promise of a practical application of SERS will be a lot closer to being fulfilled. We deeply thank the reviewer for his/her favourable comments.

We hope that this revised manuscript is now acceptable for publication in *Nature Communications*. We thank the Editorial staff and the reviewers for their help in improving our manuscript for publication.

Sincerely yours,
Hakan Usta

Corresponding authors:
Hakan Usta, Gökhan Demirel

Submitting author:

Department of Nanotechnology Engineering
Abdullah Gül University, Kayseri, 38080, Turkey
Telephone: (+90)352-2248800, Fax: (+90)352-338 88 28